# Application of the "Novel Foods" Regulation to Botanicals in the European Union

**Javier Morán [1],\* and Alina Kilasoniya [2]**

1   Department of Food Innovation, Universidad Católica San Antonio de Murcia, 30107 Murcia, Spain
2   International PhD School, Universidad Católica San Antonio de Murcia, 30107 Murcia, Spain;
    akilasoniya@alu.ucam.edu
\*   Correspondence: jmoran@sat.ucam.edu

**Abstract:** The European Union classifies "novel foods" as those not widely consumed before 15 May 1997. This category includes recently created, innovative foods, as well as those made using new technologies and processes, and foods with a traditional consumption history outside the EU. Distinguishing between "novel" and "conventional" foods is legally significant, as the former require official authorization under the Novel Foods Regulation. The regulation prioritizes safety, accurate labeling, and nutritional parity with replaced foods. Regulation (EU) 2015/2283, effective from 1 January 2018, replaced prior regulations, facilitating access to the EU market for novel and innovative foods while maintaining high safety standards. Classifying botanical products as novel can be intricate. Safety assessments for plant products must consider diversity in species, varieties, ecotypes, and chemotypes, as cultivation practices influence chemical composition. The article reviews the legislation applicable to botanicals and proposes different ways to evaluate in advance whether a product is "novel" or not, emphasizing the evaluation of the origin and consumption history of foods of plant origin.

**Keywords:** novel foods; botanicals; European regulation

## 1. Introduction

Regulation 2015/2283 (European Commission 2015) replaced, as of 31 December 2017, Regulation 259/97 (European Commission 1997a). With the entry into force of the new Regulation, particularly with the centralization of the evaluation process, it is very likely that applications for novel foods will increase and, in fact, with the previous Regulation, 125 novel foods were authorized in a period of more than 15 years while, in just the first two years of the new Regulation, EFSA received around 80 new applications for novel foods. Among the applications of novel foods, botanical preparations are one of the main areas of interest. Hence the interest in reviewing this specific aspect.

Food business operators must check for each new food or food ingredient whether it falls within the scope of the new Regulation before marketing it in the EU and must consult a national authority if they are unsure of the regulatory status. This is a legal obligation, but not a systematic requirement, so the food business operator must make a formal judgment on whether the information available to support whether or not a food falls within the scope of the new Regulation is sufficiently conclusive. In order to prove that a food ingredient is not a novel food, clear risk assessment information must be obtained including, in particular, data on the production process, composition and specifications, history of use, proposed uses and anticipated intake, absorption, distribution, metabolism and excretion, nutritional information, toxicological information, and allergenicity. This article reviews the most relevant points for making decisions before launching a new food ingredient on the EU market from the point of view of whether or not it can be considered as a novel food. Only after an "in-house" analysis of the new ingredient following the guidelines

suggested in this article and in the case of reasonable doubts about the regulatory status should the provisions established by Implementing Regulation (EU) 2018/456 (European Commission 2018) be resorted to, which establishes the phases of the consultation process to determine novel food status in accordance with the new Novel Foods Regulation.

Thus, this work aims to offer harmonized guidelines for the evaluation of new food ingredients based on botanical preparations.

European legislation on "novel foods" establishes that these are products that were not widely consumed in the European Union before 15 May 1997, the date on which the first Regulation on novel foods came into effect. This category encompasses newly created innovative foods, those produced through new technologies and processes, as well as those with a tradition of consumption outside the EU.

The regulation of these foods in the European Union is based on fundamental principles. First and foremost, they must be safe for human consumption. Additionally, they require accurate labeling that does not mislead consumers. If a "novel food" is intended to replace another, it must not have unfavorable nutritional differences for the consumer. Finally, prior authorization for marketing is required, based on a safety assessment in line with the mentioned principles.

As of 1 January 2018, Regulation (EU) 2015/2283 on novel foods came into effect, repealing and replacing Regulations (EC) No 258/97 and (EC) No 1852/2001 (European Commission 2001) that were in force until 31 December 2017. This new legal framework improves conditions for food companies to easily introduce innovative and novel foods into the EU market, without compromising the high level of food safety for European consumers. For the purposes of this article, we are interested in Article 3.2(iv) of Regulation (EU) 2015/2283: "foods consisting of, or isolated from, or produced from plants or their parts, unless the food has a history of safe food use in the Union market and consists of, or has been isolated or produced from a plant or a variety of the same species".

To facilitate the implementation of the new Regulation, the European Commission has adopted Implementing Regulation (EU) 2017/2469 (European Commission 2017), which specifies the administrative, technical, and scientific requirements that must be included in an application for a "novel food". Additionally, the European Food Safety Authority (EFSA) has issued guidance, emphasizing the importance of reporting if the production process is novel and characterizing the innovative aspects of the process (European Food Safety Authority 2016).

Regarding "significant changes" in the production process, an EFSA expert panel established in 2009 that the final product should be identical to a traditional food for which historical data is available (European Food Safety Authority 2009).

In connection with this legislation, Article 8 of Regulation (EC) No 1925/2006 (European Commission 2006) establishes the procedure for including, following an evaluation by EFSA, certain substances in its Annex III, with the aim of having better control over the addition of other substances to foods. Implementing Regulation (EU) No 307/2012 (European Commission 2012) develops the procedure provided for in that article for the inclusion of substances in Annex III.

Finally, when it cannot be determined whether a specific food falls under the category of "novel food", a consultation request must be submitted as provided for in Implementing Regulation (EU) 2018/456.

The authorization process for the marketing of a novel food in the European Union involves a series of rigorous stages to ensure safety and compliance with current regulations. These are the key phases:

1.  Establishment of Novel Food Status: To begin, a food operator must determine if the product in question was widely consumed in the European Union before 15 May 1997. In case there are no clear records of its consumption, a consultation request is required. This request is reviewed and verified by the Member States to ensure its validity. Detailed guidelines have been established in Commission Implementing Regulation (EU) 2018/456, in order to standardize this process.

2.  Demonstration of Significant Consumption History: In cases where conclusive information is not found after consulting previous records, it falls on the producer, importer, or other parties responsible for marketing in the EU market to provide the necessary information justifying that the product is not subject to the scope of Regulation (EU) 2015/2283. It is crucial to emphasize that economic operators play a crucial role in ensuring compliance with current food safety legislation.

3.  Application for New Authorization of a Novel Food (Application Procedure): Once it is confirmed that a food is classified as "novel," it must undergo a safety assessment carried out by the European Food Safety Authority (EFSA) before authorization is granted for its marketing throughout the European Union, in accordance with Regulation (EU) 2015/2283 on novel foods.

It is important to note that the new regulation on Novel Foods does not apply in certain cases, such as food additives, flavors, extraction solvents, genetically modified organisms, and certain uses of foods or ingredients in food supplements, which require authorizations under specific regulations. For example, food enrichment requires authorization under the Regulation on novel foods if it is for a use different from that in food supplements.

In summary, the authorization process for the marketing of novel foods in the European Union is a meticulous procedure that ensures safety and regulatory compliance, with the aim of safeguarding the health and well-being of European consumers.

## 2. How to Determine Whether a Product Is a "Novel Food"

A product is not considered a novel food if it has been used commercially for human consumption to a significant extent within the European Union before 15 May 1997, the date on which the Novel Food Regulation was adopted. An economic operator wishing to place a food on the market in the European Union that is not widely known is responsible for providing data demonstrating the history of consumption of the food before 1997. In case of doubt, he will submit a request for consultation to the competent authority of the Member State which will verify it in accordance with Implementing Regulation (EU) 2018/456 but, prior to this, he could assess for himself whether or not his product falls under the "novel food" classification.

### 2.1. Establishment of Novel Food Status

To determine whether a food is classified as "novel" in the European Union, it is checked whether it has been consumed on a large scale in the region before 15 May 1997, the date of entry into force of the Novel Foods Regulation. It is essential to note that consumption of the product in other parts of the world is not taken into consideration for this purpose.

An exhaustive list of authorized novel foods or products is not maintained. The responsibility falls on food operators to ensure that the products they offer to the market comply with food regulations. In the case of novel foods, operators are expected to establish and, if necessary, provide data demonstrating the consumption history of the product before 1997. This is particularly relevant for ingredients that are not widely known for their use as food.

If a dietary consumption history cannot be established, the product would be considered a novel food, subject to the scope of the Novel Foods Regulation.

The determination of novel food status is carried out in collaboration with EU food experts, based on available knowledge and materials. These interpretations at community level are recorded in the Public Catalog of Novel Foods maintained by the European Commission (Novel Food status Catalogue (New) 2024). To support the demonstration of consumption history, the Commission's Novel Foods Catalog can be used. However, it is essential to highlight that this is not a complete inventory and reflects information shared between Member States and the Commission to determine whether a product falls within the scope of the Novel Foods Regulation. The Novel Foods Catalog is dynamic and its content may evolve as new information is provided by Member States or through studies

notified to the Commission. Any correction or additional information must be submitted to the competent authorities of a Member State for verification.

In addition, it is crucial to verify if the starting product is on the lists of "traditional agri-food products", such as the Italian ones (Ministero dell'agricoltura, della sovranità alimentare e delle foreste 2023). These lists are established by the Ministry of Agricultural Policies in collaboration with the regions and are updated and published annually. They focus on products that have been processed according to ancient recipes, following methods consolidated over time.

Another document for consideration is the "EFSA Compendium" (European Commission 2024b), which identifies plants that contain substances that may raise concerns for human health when used in foods and food supplements. The EFSA published this compendium in 2012, with the purpose of facilitating the evaluation of specific ingredients in dietary supplements. It contains information on plants subject to restrictions or negative lists in at least one European Member State.

In summary, the establishment of novel food status involves a detailed assessment of its consumption history and is based on the collaboration of EU experts. The Public Catalog of Novel Foods, the Commission's Catalog of Novel Foods and other sources are used to support this determination. Lists of traditional agri-food products and the EFSA Compendium should also be considered to ensure compliance with current regulations.

### 2.2. Safe Use History

In the area of food regulations, the notion of the history of use can be defined as the body of knowledge that exists about the ways in which botanicals (and other products) and their preparations have traditionally or culturally evolved and been used over time. Typically, such usage history should cover at least one generation (25 years). A long history of use may be indicative that the botanical preparation has been used safely. A number of key parameters have been described and used to characterize the traditional use of botanicals for nutritional and physiological purposes and are summarized in the Table 1 (Anton et al. 2019; Colombo et al. 2020; van Breemen 2015).

**Table 1.** Key parameters to characterize the traditional use of botanical products for nutritional and physiological purposes (Anton et al. 2019; Colombo et al. 2020; van Breemen 2015).

| Parameter | | Details |
|---|---|---|
| Identity | | Scientific name (including author), see vernacular and common name, botanical family |
| Scope of use | | Widespread in populations/regions |
| Time/duration of use | | Over generations (e.g., ≥25–30 years) |
| Conditions of Use | Collection and treatment of botanical | Fresh, dry, etc. |
| | Botanical part | Leaf, flower, fruit, seed, root, stem, pollen, bark, tuber, rhizome, resin, exudates, etc. |
| | Mode/nature of preparation | Botanical whole or part, fresh, juice, dried, cooked, infused, extracted, distilled, etc. |
| | How to use | Orally, individually or in combinations (mixtures), etc. |
| | Usage quantity | Amount and frequency of intake |
| Use provided | | Benefit, nutritional and physiological |
| Documentation | | Authentic texts (Books, recipes, monographs, recognized textbooks, scientific and other publications |

Over time, botanical products have been employed in various ways, whether as food ingredients or for specific purposes, resulting in a wide range of preparation methods.

Many of these techniques have a long-standing tradition, some even spanning centuries, rendering them "traditional".

This concept of tradition is also acknowledged in EU drug legislation (Directive 2004/24/EC) (European Commission 2004a), where it is defined as a minimum of 30 years of use as a drug, including at least 15 years within the EU. For a product to be considered traditional, it is required to be safe under specified conditions of use, and its pharmacological effects should be plausible based on accumulated experience. This applies to individual products and preparations but does not imply a long-term evaluation.

Key parameters have been identified to characterize the traditional use of botanical products for nutritional and physiological purposes. This concept is not based on a scientific risk assessment but on the experience accumulated over generations and the absence of known adverse effects.

The nature of the various techniques used to treat and prepare botanical species is very diverse. An overview of traditional preparation methods is presented in Table 2 (Neely et al. 2011; Rietjens et al. 2008; Speijers et al. 2010; Roe et al. 2018), although preparations to be considered traditional must be evaluated on a case-by-case basis (Anton et al. 2012a).

**Table 2.** Overview of botanical preparation methods usually considered as "traditional" (Neely et al. 2011; Rietjens et al. 2008; Speijers et al. 2010; Roe et al. 2018).

| Traditional Preparation | Characteristics | |
| --- | --- | --- |
| Crushed or dried herb powder | Powders are prepared from plant material (bark, root, leaf, etc.) that is dried at a predetermined temperature. If necessary, the product is sieved to generally allow 97% of the material to pass through. | |
| Tinctures | Tinctures are liquid preparations that are generally obtained by percolation or maceration in ethanol (for example, one part plant and five parts alcohol, the alcoholic strength of which is specific for the type of plant in question) or by dissolving a dry extract in alcohol of appropriate alcoholic strength (generally 45° to 70°). | |
| Extracts | Extracts are liquid (liquid extracts and tinctures), semisolid (soft or firm extracts) or solid (dry extracts) preparations generally obtained from dried plant material. Titrated extracts are adjusted by an inert substance or by mixing batches of extracts, with an acceptable tolerance for the content of particular compounds with known physiological activity. Quantitative extracts are adjusted to a certain range of components by mixing batches of extracts. | Maceration: the raw plant material, finely distributed, is brought into contact with the total volume of solvent and kept for several days at room temperature, protected from light and shaken frequently. The mixture is removed and the remains (pomace or pomace) are pressed. The extracted solution becomes an extract after the evaporation of the solvent. |
| | | Percolation: Extraction is carried out by slowly and frequently passing the solvent through the plant powder in a percolator. The quantities of solvents and plants, the duration and flow rate of the solvent, the temperature and agitation, the granulometry of the plant, and the type of apparatus used are parameters that can influence the extraction. |
| Liquid extracts | Liquid extracts are preparations in which, in general, one part by weight or volume corresponds to one part by weight of dry plant. These preparations are adjusted, if necessary, to achieve the desired level of components. The advantage is that 1 g of liquid extract corresponds to 1 g of dry plant. The extraction process is carried out with ethanol of adequate strength or with water. | |
| Liquid Glycerin Extracts | These extracts are generally obtained from freeze-dried or ground fresh plant material, to avoid the degradation of the active compounds. A first extraction with a mixture of water and alcohol of increasing concentration allows the water-soluble and fat-soluble compounds to be obtained. The solvent is removed in vacuo and the extract is resuspended in a glycerin solution (15% water and 85% glycerin). This form is the one that most closely approximates the chemical profile of the fresco. | |

**Table 2.** *Cont.*

| Traditional Preparation | Characteristics |
|---|---|
| Soft or firm extracts | Soft and firm extracts are semisolid preparations obtained by the evaporation or partial evaporation of the material used for the production of the extract. |
| Dry extracts | Dry extracts are solid preparations obtained by the evaporation of the solvent with which the extract was produced. The solvent can be water (aqueous extract) or alcohol, whose graduation is important (hydroalcoholic extract). |
| Essential oils | The essential oil can be obtained by the steam (water) treatment of dried or fresh plant material, possibly followed by rectification, and by the mechanical pressing of the fresh pericarp (citrus peel) without applying heat. Steam hydrodistillation gives rise to two fractions: a lipophilic fraction (the essential oil), which generally floats (in most cases the density is less than 1) and an aqueous fraction, which is called "distilled water" from plant. Obviously, the components of both fractions are not the same. It is also possible to extract compounds using supercritical C02, but the product obtained cannot be called "essential oil". |
| Oleoresin and gum-oleoresin | An oleoresin is a natural secretion of a plant, obtained by incisions in the bark of the tree, like the exudates of conifers. It consists of a mixture of oxidized essential oil and resins. Other gums, resins or gum-resins, are mixtures of complex resinous compounds, gums and volatile compounds. |
| Balms | Balms are specific natural oleoresins characterized by the presence of benzoic acid and/or cinnamic acid and their esters. Many preparations are incorrectly called balm. |
| Alcoholate | These preparations are liquid solutions obtained by the distillation of generally fresh plant material in ethanol. Alcoholates can be obtained from a single plant (simple alcoholate) or from a mixture of several plants (mixed alcoholate). |
| Alcohol | This preparation is obtained by the maceration of fresh plant material in highly concentrated cold ethanol to preserve compounds that would lose their activity if they dried. |
| Intract | This is a plant extract obtained after treating the plant using heat or alcoholic vapors in order to deactivate the enzymes present in the plant. |
| Macerated | There are two types of maceration: Cold maceration applied to all parts of the plants rich in mucilage and hot maceration used for the hard parts of the plants (root, wood). The mash can be aqueous or oily, depending on the type of substances (hydrophilic or lipophilic). It is generally applied to dry plant material to avoid the presence of water naturally present in the plant. |
| Mellitum | This is a complex and viscous liquid vegetable preparation in which honey is a main ingredient along with an aqueous component (water, vinegar, etc.). |

The use of extraction solvents is also relevant, as these are substances capable of dissolving botanical compounds and contaminants. The choice of solvents is based on their physico-chemical properties that enable the concentration of compounds during manufacturing. Directive 2009/32/EC (European Commission 2009) and its implementation regulate authorized extraction solvents and associated processes. Permitted solvents include propane, butane, ethyl acetate, ethanol, carbon dioxide, acetone, and nitrous oxide.

The information regarding the historical use of a botanical is typically broad in scope, but it must encompass crucial details that define the parameters of its traditional usage. This encompasses specifics about the plant, the specific part used, the manner of preparation (including details like method and type of solvent employed), and the customary range of quantities utilized in its preparation. In certain instances, additional aspects of traditional use must be specified, such as the plant's region of origin, harvest timing, drying method, extraction, and other relevant processes. A more comprehensive availability of information enhances the ability to discern the traditional pathway.

If a preparation crafted by a manufacturer aligns with the general attributes of traditional use, it can be categorized as traditional. It is noteworthy that current technologies permit the examination of the composition of a traditional preparation, enabling an assessment of whether the manufactured version is equivalent. The greater the deviation

of a preparation from the traditional one, the less dependable the tradition of use is in supporting its purported benefits or safety. This necessitates a more thorough evaluation of its benefits and safety.

When key elements of a preparation differ, such as the plant part used, the preparation technique, or the type of solvent, the resulting product cannot be considered on par with the traditional one. If the preparation method is akin but certain process parameters are altered (e.g., alcohol concentration, duration, temperature), the manufacturer must evaluate how these variations impact the chemical profile of the resultant product. If it can be demonstrated that the chemical profile aligns with the traditional one, the preparation may be regarded as adhering to traditional use (Anton et al. 2012b).

There are three scenarios to consider:

1. When the specific compounds responsible for the benefits or safety concerns are identified, their presence and quantity in the preparation should match those in the traditional one. In this case, it can be demonstrated that the preparations faithfully represent the traditional preparation, validating the associated benefits and safety.
2. If the beneficial effects or safety concerns are associated with compounds from a well-known chemical group, the chemical profile of this group should be akin, both qualitatively and quantitatively, to that of the traditional preparation. Depending on the chemical group in question, preparations can be standardized or quantified.
3. In cases where the compounds responsible for the beneficial effects or safety concerns are unidentified, the chemical composition of the preparation must mirror that of the traditional one for traditional knowledge about benefits and safety to be applicable.

While these scenarios address the use of traditional knowledge for benefit assessment, it is important to note that the safety characteristics of preparations may still differ and should be assessed based on the comprehensive chemical profile of the preparation. Furthermore, cross-referencing with the EFSA compendium listing compounds of concern also allows manufacturers to verify whether compounds of concern are relevant to the traditional preparation and, where appropriate, apply quality assurance measures to control them.

In the EU, there is a pre-market authorization process for certain food categories that were not significantly used in the EU before 15 May 1997. This includes foods and food ingredients derived from plants, except those obtained through traditional methods with a history of safe food use. Botanicals meeting these criteria are not subject to the new procedure. Although many of these botanicals are listed in the EU novel foods catalogue, applying conventional and traditional processing methods does not classify the extract or final product as novel foods in member states. However, finding documentation on the use of botanicals before 1997 can be challenging due to limited electronic records and most information being discarded by manufacturers. Therefore, additional sources of information are valuable in confirming the traditional nature of the botanical product and its distribution. General information about the traditional nature of preparations or products can be found and can help affirm the history of safe use and identify methods traditionally applied to specific botanicals. Botany textbooks and monographs are often rich sources of such knowledge (Lenssen et al. 2022).

Another category of novel foods encompasses foods and food ingredients subjected to a production process not currently in use. This process induces substantial alterations in the composition or structure of the food or its ingredients, impacting their nutritional content, metabolism, or presence of undesirable substances. Essentially, this implies that the processing itself can categorize a preparation as a novel food (process) if it yields products notably distinct from their conventional or traditional counterparts. Notably, this does not apply when employing traditional processing methods for botanical products, resulting in traditional preparations as previously outlined. Additionally, if process parameters are adjusted but the product still aligns with the traditional preparation, being akin to it in significant aspects, it can be considered equivalent, although the effects of alterations must be thoroughly assessed and documented. Conversely, if the fundamental attributes of the

preparation are changed, markedly different preparations can be derived. In such instances, the extent of disparity must be evaluated individually to determine whether modifications in the food or ingredient's composition or structure, affecting its nutritional value, metabolism, or undesirable substance levels, are substantial and warrant classification under novel foods legislation (European Food Safety Authority 2012).

In summary, safe use history is based on accumulated experience and the absence of known adverse effects. Extraction solvents are assessed based on their ability to dissolve botanical compounds. Additionally, the tradition of use can be supported with information about botanical preparations and their benefits from monographs and botanical books. This confirms the importance of tradition in the safety and effectiveness of botanical products used in foods and medicines.

### 2.3. Concentration Methods of Bioactive Ingredients

In Regulation 2283/2015, the concept of "produced from" was introduced without detailed explanation. This term broadly encompasses foods derived from plant material, which constitutes the majority of foods and food ingredients in the EU. However, it is important to clarify that it is not intended to encompass all new foods, but primarily pertains to plant-derived products that have not been previously marketed. Under the original Regulation 258/97, very few foods derived directly from plants or their parts (without isolation) underwent the novel food authorization process. Extracts from plants with established use prior to 15 May 1997 typically fall outside the scope. Additionally, the novel foods catalog primarily addresses the status of plant species rather than all products stemming from them, a trend unlikely to shift. Nevertheless, foods or ingredients derived from plants, which, while not isolated, substantially differ from the original plant (e.g., specific preparations or highly selective extracts), may be subject to the new Regulation. Again, in cases where plants or plant parts lacking a history of safe use before 15 May 1997 serve as source material for food production, the resulting food falls under the new Regulation.

The Italian authorities have confirmed that plant preparations and extracts, as per the decree of 10 August 2018 (Ministero della Salute 2022), must have levels of active substances comparable to traditional methods to avoid falling under the Novel Foods Regulation.

When utilizing plants or plant parts with established safe usage, the likelihood of the food having a history of safe use is higher. However, it is crucial to separately assess the specific processing methods employed. Demonstrating the history of safe use in such cases remains a key question. Food business operators can consider various factors to confirm that their product does not fall under the novel foods category:

1.  Market History: The regulation maintains 15 May 1997, as the cutoff for determining meaningful use. However, due to the inclusion of the term "produced from," it could potentially encompass numerous foods from plants or plant parts on the market between 15 May 1997, and 1 January 2018. Nonetheless, products legally on the market should generally not face retroactive challenges, especially those properly notified to national authorities in compliance with national and EU regulations.
2.  Nature of the Product: The degree of processing can significantly alter foods from their source material. Foods closer in characteristics to the source material or a traditionally used food from the same source material are more likely to retain the safe use history. This is particularly relevant for isolated foods, which retain fewer of these characteristics, while other preparations and extracts may preserve more.

Demonstrating that a product has been on the market prior to 15 May 1997 is the strongest evidence of its history of safe food use. Alternatively, showing similarity to products marketed before this date may also be acceptable. Evidence can be sourced from various channels. In the case of botanical preparations and extracts, knowledge of traditional plant material processing is valuable in justifying safe food use. It is important to note that demonstrating a history of safe food use is not synonymous with demonstrating significant use before 15 May 1997.

According to Article 1(2) of Regulation (EC) No 258/97, novel foods and novel food ingredients are foods that have not been used to a significant extent for human consumption in the community before the entry into force of the Regulations (15 May 1997). In certain cases, it may be difficult to determine whether this particular "significant consumption" provision is met and that is why the European Commission produced a document (European Commission 2004b) that aims to help competent authorities and interested parties to better understand and apply the concept correctly and uniformly, although it does not provide either a list or an exhaustive classification of the relevant criteria that must be considered since, in its opinion, each product must be evaluated on a case-by-case basis.

The concept of "substantial equivalence" offers a valuable framework for evaluating how a new plant preparation or extract compares to existing counterparts with a well-established history of safe food use. Originally introduced by the WHO (Food Safety Team 1995) and OECD (Organisation for Economic Co-operation and Development 1993), this concept provides a method for assessing the safety of a potential new food in relation to its conventional counterpart. If a novel food is deemed to be substantially equivalent, it can be handled in a similar manner with regards to safety considerations. However, it is important to underscore that substantial equivalence does not replace the need for compliance with novel foods legislation, especially for products falling within the new Regulation's definition. This principle should solely be utilized to gauge the extent of differentiation from foods with a history of use prior to 15 May 1997. It cannot be utilized to validate a non-novel status for foods failing to meet the specifications or conditions of already authorized novel foods. Food business operators bear the responsibility of ensuring the safety of their products. Therefore, the elements discussed in this context should be meticulously addressed and maintained as justification for the verification's outcome, particularly if premarket authorization is not mandated under the new Regulation.

The aspects that must be taken into account when judging the extent of the changes that are acceptable for a food to be considered novel are discussed below:

Choosing a Reference Food: Substantial equivalence necessitates comparing all aspects of the food with a similar reference food and the reference food should have a history of significant use in the EU before 15 May 1997, or be an already approved novel food. The closer the reference food is to the new (novel) food, the more meaningful the comparison.

Nutritional and Non-Nutritional Composition: Comprehensive information on contaminants, residues, and other naturally occurring substances that may affect safety is essential and a detailed literature search should identify any relevant substances associated with the food and its origin.

The status of "selective extracts" is crucial in the new Regulation. It is clarified that foods made exclusively from non-regulated food ingredients are not considered novel. However, if a food ingredient has not been widely consumed in the EU, it falls under the Regulation. The focus is on products modified from ingredients without a history of safe use. The intent is not to cover all new extracts, but only those altered from EU-unsupported ingredients. This aligns with historical Regulation application, where novel foods primarily involved isolated compounds. Plant extracts entered the process when substantially modified. While each extract has some selectivity, established-use and entirely new extracts exist. When compounds are selectively enriched, a robust use history justification is needed. As long as typical source material characteristics are retained in the preparation, the source material's safety applies. However, in many extracts, assessing properties through taste may be unfeasible. In such cases, an extract analysis is necessary. Generally, if the relative proportion of components in the extract mirrors that in the source material, it is not deemed selective. Accredited lab values are essential to show the relative compound proportion remains largely unaltered. This also applies to demonstrating that processing does not cause significant changes.

1. Novel food authorizations: An authorized selective extract, Glavonoid®, is derived from licorice root (*Glycyrrhiza glabra* L.) using ethanol and medium chain triglycerides (MCT). It contains 30% ethanolic licorice root extract and 70% MCT, with a reported

24% polyphenol content. This process significantly alters the plant's profile by concentrating hydrophobic polyphenolic substances and reducing glycyrrhizinic acid below 0.005% (*w/w*) (European Commission 2011a).

2. Notable additions to the Catalog of Novel Foods have emerged from Member State discussions: Griffonia seeds simplicifolia, used exclusively in food supplements before 15 May 1997, are considered a novel food, potentially subject to national restrictions (March 2022). Curcuminoids from turmeric extract (*Curcuma longa* L.) used exclusively in food supplements before 15 May 1997, may contain various curcuminoids. Any process enhancing solubility or bioavailability could be subject to novel food regulations (since June 2022).

Relative Ratio of Compounds: Absolute content of certain compounds is not the primary parameter; the relative ratio is key. Levels in a final product are governed by EU or national legislation, ensuring safety.

Primary Production: Only products directly derived from plants or animals qualify. Traditional food processing techniques can be applied, but compounds isolated or produced from edible parts may not qualify.

Specific Examples: Examples provided demonstrate scenarios like selective extraction, traditional processing methods and specific physical states of consumption.

Interpreting Novelty: Novelty is assessed based on significant changes involving the ingestion of traditionally non-food molecular structures.

This summary covers the key points related to judging the extent of changes acceptable for a food to be considered novel and to specify the "novel" character of a product, a series of categories have been established to which a new food must belong:

1. Constitute or be derived from a new (novel) source: that is, a new animal, plant or microbiological species.
2. Be derived from a new (novel) process: that is, the use of new (novel) technologies (in the field of food) on conventional food sources, could, just because they are not known, structurally modify the sources giving rise to some possible risk for the consumer.

An additional aspect to consider is established from the conclusions published in the Consultation process on novel food status (European Commission 2024a). We can verify that these opinions focus on several fundamental aspects:

1. In the case of food: If the food and the manufacturing process were used in the EU before 14 May 1997.
2. In the case of food supplements: If there is national legislation that authorizes their use as long as the manufacturing process is not new (novel).
3. The traditional (not novel) manufacturing processes that should be part of those included in the "List of traditional food preparation procedures" of Annex II of Regulation No. 1334/2008 (European Commission 2008), which includes the provisions of Section 4 of the Recommendation. 97/618/EC of the Commission (European Commission 1997b) (Class 6: "Food produced by a new process" which includes foods and food ingredients that have been subjected to a process not traditionally used in food production and that therefore fall within the scope of application of novel foods legislation which indicates that the resulting product is considered a novel food if the process causes a modification of the structure or chemical composition of the food ingredient that affects its nutritional value, metabolism or level of undesirable substances).
4. The use of extraction solvents in relation to Directive 2009/32/EC.
5. The definition of "partially" processed food established in Article 2 of Regulation No. 178/2002 (European Commission 2002) (For the purposes of this Regulation, "food" (or "food product") shall be understood as any substance or product intended to be ingested by man or with a reasonable probability of being so, whether or not it has been totally or partially transformed.

6.  Further explanation of "significant" consumption of a food as any food that was not used to a large extent for human consumption in the Union before 15 May 1997, regardless of the accession dates of the Member States to the Union and that falls into at least one of the categories of the Novel Foods Regulation.

*2.4. Nanomaterials*

In January 2018, the European Union implemented Regulation (EU) 2015/2283, introducing significant changes to the process of approving novel foods. Notably, since 2018, the definition of "novel" also encompasses engineered nanomaterials.

This inclusion of "nanomaterial engineering" in the definition of novel foods signifies a clear acknowledgment of advancements in food technology. Nanomaterial engineering involves the deliberate manipulation of food particles to impart specific benefits, such as enhanced nutritional content or reduced fat content. Interestingly, the definition has not been revised in light of the Commission's 2011 Recommendation (European Commission 2011c), which proposed criteria for classifying materials containing nanomaterials. According to this recommendation, a material would not be considered to contain nanomaterials if 50% or less of its particles fell within the range of 1 to 100 nanometers. However, in cases where environmental, health, safety, or competitiveness concerns arise, this threshold could potentially be further reduced by 1 to 50%.

These developments underscore the EU's commitment to ensuring the safety and integrity of novel foods, especially those incorporating cutting-edge technologies like nanomaterial engineering. This regulatory framework serves as a vital foundation for the future of food innovation and its integration into the European market.

**3. Conclusions**

Since the new Novel Foods Regulation came into force, several articles have been published on the scope of the new regulation on the placing of new ingredients on the market in the European Union (Vettorazzi et al. 2020; Ververis et al. 2020; Garcia-Vello et al. 2022; Komala et al. 2023). However, the situation of botanicals has hardly been addressed, which is why we decided to work on this issue that is so important since most of the new functional foods and food supplements that have been launched recently on the European market contain botanicals especially to take advantage of the use of "pending" statements on botanicals (European Food Safety Authority 2023; European Commission 2011b).

In the European Union, "novel foods" are defined as those that were not consumed to a significant extent before 15 May 1997, when the first Regulation on Novel Foods came into effect. This category encompasses recently created, innovative foods, as well as those produced using new technologies and processes, and foods with a traditional consumption history outside the EU.

It is crucial to differentiate between "novel" and "conventional" foods in legal terms in Europe, as the former require an official authorization for marketing under the Novel Foods Regulation. The regulation of novel foods in the EU is based on several fundamental principles. Firstly, they must be safe for consumers. Additionally, they must bear accurate labeling to prevent confusion among consumers. If a novel food is intended to replace another, it must not be nutritionally disadvantageous compared to the food it is replacing. Finally, prior authorization for the marketing of novel foods is required, based on a safety assessment that complies with the aforementioned criteria.

As of 1 January 2018, Regulation (EU) 2015/2283 on Novel Foods was implemented, repealing and replacing the previous Regulations (EC) No. 258/97 and (EC) No. 1852/2001, which were in force until 31 December 2017. This new Regulation facilitates access for food businesses to the EU market with novel and innovative foods, while maintaining a high standard of food safety.

To streamline the enforcement of the new Regulation, the Commission has established specific requirements in Implementing Regulation (EU) 2017/2469. This regulatory act outlines the administrative, technical, and scientific aspects that must be included in an

application for a novel food. Additionally, the European Food Safety Authority (EFSA) has issued a guide specifying the details of the production process.

There are cases, particularly in botanical contexts, where classifying a product as a novel food can be confusing. To determine novelty, categories have been established, including the origin of a new food source or the use of a new process in the production of conventional foods, which may imply potential risks for the consumer.

The safety assessment of plant products must take into account diversity in terms of species, varieties, ecotypes, and chemotypes. This is because factors such as cultivation practices can influence the chemical composition of products, even when plants are botanically similar.

This article pays special attention to foods consisting of plants, their parts, or derivatives, analyzing potential approaches to initially determine if such a food has a safe consumption history in the European Union and originates from a plant or variety of the same species, or has been isolated or produced from it, as in these cases it would not be considered "novel".

**Funding:** The writing and publication of this article was funded by Food Consulting & Associates.

**Institutional Review Board Statement:** Not applicable.

**Informed Consent Statement:** Not applicable.

**Data Availability Statement:** The data presented in this study are available on request.

**Conflicts of Interest:** The authors declare that they have no conflict of interest.

## References and Notes

European Commission. 1997a. Regulation (EC) No 258/97 of the European Parliament and of the Council of 27 January 1997 on novel foods and novel food ingredients.

European Commission. 2001. Regulation (EC) No 1852/2001 of 20 September 2001 laying down detailed rules for making certain information public and for the protection of information provided in accordance with Regulation (EC) No 258/97 of the European Parliament and of the Council.

European Commission. 2006. Regulation (EC) No 1925/2006 of the European Parliament and of the Council of 20 December 2006 on the addition of vitamins, minerals and certain other substances to foods.

European Commission. 2011a. 2011/761/EU: Commission Implementing Decision of 24 November 2011 authorizing the placing on the market of flavonoids from Glycyrrhiza glabra L. as a novel food ingredient, in accordance with Regulation (EC) No 258/ 97 of the European Parliament and of the Council [notified under number C(2011) 8362].

European Commission. 2012. Implementing Regulation (EU) No 307/2012 of 11 April 2012 laying down detailed rules for the application of Article 8 of Regulation (EC) No 1925/2006 of the European Parliament and of the Council, on the addition of vitamins and minerals and certain other substances to foods.

European Commission. 2015. Regulation (EU) 2015/2283 of the European Parliament and of the Council of 25 November 2015 on novel foods, amending Regulation (EU) No 1169/2011 of the European Parliament and of the Council and amending repeal Regulation (EC) No 258/97 of the European Parliament and of the Council and Regulation (EC) No 1852/2001 of the Commission.

European Commission. 2017. Commission Implementing Regulation (EU) 2017/2469 of 20 December 2017 laying down the administrative and scientific requirements to be met by applications referred to in Article 10 of Parliament Regulation (EU) 2015/2283 European and Council, relating to novel foods.

European Commission. 2018. Regulation (EU) 2018/456 of 19 March 2018 on the phases of the consultation process to determine novel food status in accordance with Regulation (EU) 2015/2283 of the European Parliament and of the Council, related to novel foods.

European Food Safety Authority. 2016. EFSA NDA Panel (EFSA Panel on Dietetic Products, Nutrition and Allergies), Dominique Turck, Jean-Louis Bresson, Barbara Burlingame, Tara Dean, Susan Fairweather-Tait, Marina Heinonen, Karen Ildico Hirsch-Ernst, Inge Mangelsdorf, Harry McArdle, and et al. 2016. Guidance on the preparation and presentation of an application for authorization of a novel food in the context of Regulation (EU) 2015/2283. *EFSA Journal* 14: 4594.

Novel Food status Catalogue (New). 2024. Available online: https://food.ec.europa.eu/safety/novel-food/novel-food-status-catalogue_en (accessed on 22 January 2024).

Anton, Robert, Basil Mathioudakis, Suwijiyo Pramono, Ekrem Sezik, and Surinder Sharma. 2019. Traditional Use of Botanicals and Botanical Preparations. *European Food and Feed Law Review* 14: 132–41.

Anton, Robert, Mauro Serafini, and Luc Delmulle. 2012a. Traditional Knowledge for the Assessment of Health Effects for Botanicals—A Framework for Data Collection. *European Food and Feed Law Review* 7: 241–50.

Anton, Robert, Mauro Serafini, and Luc Delmulle. 2012b. The role of Traditional Knowledge in the Safety Assessment of Botanical Food Supplements—Requirements for Manufacturers. *European Food and Feed Law Review* 5: 241–50.

Colombo, Francesca, Patrizia Restani, Simone Biella, and Chiara Di Lorenzo. 2020. Botanicals in Functional Foods and Food Supplements: Tradition, Efficacy and Regulatory Aspects. *Applied Sciences* 10: 2387. [CrossRef]

European Commission. 1997b. 97/618/EC: Commission Recommendation of 29 July 1997 relating to scientific aspects and the presentation of the information necessary to support applications for the placing on the market of novel foods and new food ingredients, the presentation of said information and the preparation of initial assessment reports in accordance with Regulation (EC) No 258/97 of the European Parliament and of the Council.

European Commission. 2002. Regulation (EC) No 178/2002 of the European Parliament and of the Council of 28 January 2002 laying down the general principles and requirements of food law, establishing the European Food Safety Authority and laying down procedures related to food safety.

European Commission. 2004a. Directive 2004/24/EC of the European Parliament and of the Council of March 31, 2004 amending, as regards traditional herbal medicinal products, Directive 2001/83/EC establishing a Community code on medicinal products for human use.

European Commission. 2004b. Human Consumption to a Significant Degree. Information and Guidance Document. Brussels.

European Commission. 2008. Regulation (EC) No 1334/2008 of the European Parliament and of the Council of 16 December 2008 on flavorings and certain food ingredients with flavoring properties used in foods and amending Regulation (EEC) No 1601/91 of the Council, Regulations (EC) No 2232/96 and (EC) No 110/2008 and Directive 2000/13/EC.

European Commission. 2009. Directive 2009/32/EC of the European Parliament and of the Council, of 23 April 2009, on the approximation of the laws of the Member States on extraction solvents used in the manufacture of foodstuffs and their ingredients.

European Commission. 2011b. Questions and Answers on the List of permitted Health Claims. Available online: https://ec.europa.eu/commission/presscorner/detail/en/MEMO_11_868 (accessed on 22 January 2024).

European Commission. 2011c. Commission Recommendation of 18 October 2011 on the definition of nanomaterial. *Official Journal of the European Union* 275: 38–40.

European Commission. 2024a. Consultation Process on Novel Food Status. Available online: https://food.ec.europa.eu/safety/novel-food/consultation-process-novel-food-status_en (accessed on 22 January 2024).

European Commission. 2024b. Botanical Summary Report. Available online: https://www.efsa.europa.eu/en/microstrategy/botanical-summary-report (accessed on 22 January 2024).

European Food Safety Authority. 2009. *EFSA Scientific Colloquium Summary Report. What's New on Novel Foods*. Amsterdam: EFSA, November 19–20.

European Food Safety Authority. 2012. Compendium of botanicals reported to contain naturally occurring substances of possible concern for human health when used in food and food supplements. *EFSA Journal* 10: 2663.

European Food Safety Authority. 2023. Botanicals. Available online: https://www.efsa.europa.eu/en/topics/topic/botanicals (accessed on 22 January 2024).

Food Safety Team. 1995. *Workshop on the Application of the Principles of Substantial Equivalence to the Safety Evaluation of Food or Food Components from Plants Derived by Modern Biotechnology*. Copenhagen: World Health Organization.

Garcia-Vello, Pilar, Kiara Aiello, Nicola M. Smith, Julia Fabrega, Konstantinos Paraskevopoulos, Marta Hugas, and Claudia Heppner. 2022. Preparing for future challenges in risk assessment in the European Union. *Trends in Biotechnology* 40: 1137–40. [CrossRef] [PubMed]

Komala, Muralikrishna Gangadharan, Ser Gin Ong, Muhammad Uzair Qadri, Lamees M. Elshafie, Carol A. Pollock, and Sonia Saad. 2023. Investigating the Regulatory Process, Safety, Efficacy and Product Transparency for Nutraceuticals in the USA, Europe and Australia. *Foods* 12: 427. [CrossRef] [PubMed]

Lenssen, Karin G. M., Aalt Bast, and Alie De Boer. 2022. The complexity of proving health effects with data on 'traditional use': A critical perspective on supporting botanical health claims. *Trends in Food Science & Technology Volume* 120: 338–43.

Ministero dell'agricoltura, della sovranità alimentare e delle foreste. 2023. Ventitreesima Revisione dell'elenco dei prodotti Agroalimentari Tradizionali. Available online: https://www.politicheagricole.it/flex/cm/pages/ServeBLOB.php/L/IT/IDPagina/19693 (accessed on 22 January 2024).

Ministero della Salute. 2022. Circolare ministeriale "Indicazioni sull'uso delle piante e delle loro parti negli integratori alimentari per garantire la sicurezza e tutela dei cittadini". 27 Maggio 2022.

Neely, Theresa, Brian Walsh-Mason, Paul Russell, Anthony Van der Horst, Sue O'Hagan, and Praful Lahorkar. 2011. A multi-criteria decision analysis model to assess the safety of botanicals utilizing data on history of use. *Toxicology International* 18: S20–S29. [CrossRef] [PubMed]

Organisation for Economic Co-operation and Development. 1993. *Safety Evaluation of Foods Derived by Modern Biotechnology: Concepts and Principles*. Paris: Organisation for Economic Co-operation and Development (OECD).

Rietjens, Ivonne MCM, Wout Slob, Corrado Galli, and Vittorio Silano. 2008. Risk assessment of botanicals and botanical preparations intended for use in food and food supplements: Emerging issues. *Toxicology Letters* 180: 131–36. [CrossRef] [PubMed]

Roe, Amy L., Donna A. McMillan, and Catherine Mahony. 2018. A Tiered Approach for the Evaluation of the Safety of Botanicals Used as Dietary Supplements: An Industry Strategy. *Clinical Pharmacology & Therapeutics* 104: 446–57.

Speijers, Gerrit, Bernard Bottex, Birgit Dusemund, Andrea Lugasi, Jaroslav Tóth, Judith Amberg-Müller, Corrado L. Galli, Vittorio Silano, and Ivonne MCM Rietjens. 2010. Safety assessment of botanicals and botanical preparations used as ingredients in food supplements: Testing an European Food Safety Authority-tiered approach. *Molecular Nutrition & Food Research* 54: 175–85.

van Breemen, Richard B. 2015. Development of Safe and Effective Botanical Dietary Supplements. *Journal of Medicinal Chemistry* 58: 8360–72. [CrossRef] [PubMed]

Ververis, Ermolaos, Reinhard Ackerl, Domenico Azzollini, Paolo Angelo Colombo, Agnès de Sesmaisons, Céline Dumas, Antonio Fernandez-Dumont, Lucien Ferreira da Costa, Andrea Germini, Tilemachos Goumperis, and et al. 2020. Novel foods in the European Union: Scientific requirements and challenges of the risk assessment process by the European Food Safety Authority. *Food Research International* 137: 109515. [CrossRef] [PubMed]

Vettorazzi, Ariane, Adela López de Cerain, Julen Sanz-Serrano, Ana G. Gil, and Amaya Azqueta. 2020. European Regulatory Framework and Safety Assessment of Food-Related Bioactive Compounds. *Nutrients* 12: 613. [CrossRef] [PubMed]

