# Peer review of "Application of the “Novel Foods” Regulation to Botanicals in the European Union"

_laws, 2023_

Round 1

Reviewer 1 Report

Comments and Suggestions for Authors

The article lacks a clear statement of purpose (which should be given in the introduction to the article). A reader can only guess from the abstract that the article refers to “legislation applicable to botanicals and proposes different ways to evaluate in advance whether a product is "novel" or not, emphasizing the evaluation of the origin and consumption history of foods of plant origin”. However, even that statement is not clear enough to clarify the purpose of the article. The article should give the reader clear information about what the authors' intention was, what they want to demonstrate and prove, and why it is important to propose different ways to evaluate in advance whether a product is "novel" or not, emphasizing the evaluation of the origin and consumption history of foods of plant origin”. Are these ways better / more sufficient than the ones applied currently? What is the hypothesis or the justification of proposing those different ways? To what extent those “different ways of evaluation” are relevant in particular for botanicals?

It is necessary to indicate by the titles of tables (no. 1 and no. 2) whether these were prepared on the basis of the authors' own sources. 

  • Generally the article is relevant for the field. There are not many cited recent publications, maybe the authors could find some recent and relevant publications referring to botanicals as a novel food in particular.
  • • The manuscript is scientifically sound, but the lack of a clear purpose and hypothesis makes it unclear what the rationale for conducting the research is. The conclusions are not detailed enough and do not sufficiently address the arguments and analyses presented in the article.

The article fits the journal scope and the subject of the article is original. The analysis of the legislation is conducted appropriately with high standards of presentation. Conclusions seem interesting for the readership (apart from the shortcomings mentioned above) and the paper has a potential to attract a wide readership. In my opinion, the work advances the current knowledge. The English language is appropriate and understandable.

Overall Recommendation

Accept after Minor Revisions: The paper can in principle be accepted after revision based on the reviewer’s comments.

Author Response

AUTHOR'S REPLY TO THE REVIEW REPORT (REVIEWER 1)

WE HAVE INCLUDED THE PURPOSE OF THE ARTICLE AT THE BEGINNING OF THE ARTICLE:

Regulation 2015/2283 replaced, as of December 31, 2017, Regulation 259/97. With the entry into force of the new Regulation, particularly with the centralization of the evaluation process, it is very likely that applications for novel foods will increase and, in fact, with the previous Regulation, 125 novel foods were authorized in a period of more than 15 years while, in just the first two years of the new Regulation, EFSA received around 80 new applications for novel foods. Among the applications of novel foods, botanical preparations are one of the main areas of interest. Hence the interest in reviewing this specific aspect.

Food business operators must check for each new food or food ingredient whether it falls within the scope of the new Regulation before marketing it in the EU and must consult a national authority if they are unsure of the regulatory status. This is a legal obligation, but not a systematic requirement, so the food business operator must make a formal judgment on whether the information available to support whether or not a food falls within the scope of the new Regulation is sufficiently conclusive. In order to prove that a food ingredient is not a novel food, clear risk assessment information must be obtained and in particular on the production process, composition data and specifications, history of use, proposed uses and anticipated intake, absorption, distribution, metabolism and excretion, nutritional information, toxicological information and allergenicity. This article reviews the most relevant points for making decisions before launching a new food ingredient on the EU market from the point of view of whether or not it can be considered a novel food. Only after an “in-house” analysis of the new ingredient following the guidelines suggested in this article and in case of reasonable doubts about the regulatory status should the provisions established by Implementing Regulation (EU) 2018/456 be resorted to, which establishes the phases of the consultation process to determine novel food status in accordance with the new Novel Foods Regulation.

Thus, this work aims to offer harmonized guidelines for the evaluation of new food ingredients based on botanical preparations.

It is necessary to indicate through the titles of the tables (No. 1 and No. 2) whether they were prepared on the basis of the authors' own sources.

THE SUPPORTING BIBLIOGRAPHY FOR TABLES 1 AND 2 HAS BEEN INCLUDED:

  1. Anton R, Mathioudakis B, Pramono S, Sezik E, Sharma S. Traditional Use of Botanicals and Botanical Preparations. European Food and Feed Law Review 2019, 14 (2), 132-141.
  2. Colombo F, Restani P, Biella S, Di Lorenzo C. Botanicals in Functional Foods and Food Supplements: Tradition, Efficacy and Regulatory Aspects. Applied Sciences. 2020; 10(7):2387.
  3. van Breemen RB. Development of Safe and Effective Botanical Dietary Supplements. J Med Chem. 2015 Nov 12;58(21):8360-72.

  1. Neely T, Walsh-Mason B, Russell P, Horst AV, O'Hagan S, Lahorkar P. A multi-criteria decision analysis model to assess the safety of botanicals utilizing data on history of use. Toxicol Int. 2011 Aug;18(Suppl 1):S20-9.
  2. Rietjens IM, Slob W, Galli C, Silano V. Risk assessment of botanicals and botanical preparations intended for use in food and food supplements: emerging issues. Toxicol Lett. 2008 Aug 15;180(2):131-6.
  3. Speijers G, Bottex B, Dusemund B, Lugasi A, Tóth J, Amberg-Müller J, Galli CL, Silano V, Rietjens IM. Safety assessment of botanicals and botanical preparations used as ingredients in food supplements: testing an European Food Safety Authority-tiered approach. Mol Nutr Food Res. 2010 Feb;54(2):175-85.
  4. Roe AL, McMillan DA, Mahony C. A Tiered Approach for the Evaluation of the Safety of Botanicals Used as Dietary Supplements: An Industry Strategy. Clin Pharmacol Ther. 2018 Sep;104(3):446-457.

NEW RECENT BIBLIOGRAPHY ON THE TOPIC HAS BEEN ADDED

Since the new Novel Foods Regulation came into force, several articles have been published on the scope of the new regulation on the placing of new ingredients on the market in the European Union. However, the situation of botanicals has hardly been addressed, which is why we decided to work on this issue that is so important since most of the new functional foods and food supplements that have been launched recently on the European market contain botanicals especially to take advantage of the use of “pending” statements on botanicals.

  1. Vettorazzi A, López de Cerain A, Sanz-Serrano J, Gil AG, Azqueta A. European Regulatory Framework and Safety Assessment of Food-Related Bioactive Compounds. Nutrients. 2020 Feb 26;12(3):613.
  2. Ververis E, Ackerl R, Azzollini D, Colombo PA, de Sesmaisons A, Dumas C, Fernandez-Dumont A, Ferreira da Costa L, Germini A, Goumperis T, Kouloura E, Matijevic L, Precup G, Roldan-Torres R, Rossi A, Svejstil R, Turla E, Gelbmann W. Novel foods in the European Union: Scientific requirements and challenges of the risk assessment process by the European Food Safety Authority. Food Res Int. 2020 Nov;137:109515.
  3. Garcia-Vello P, Aiello K, Smith NM, Fabrega J, Paraskevopoulos K, Hugas M, Heppner C. Preparing for future challenges in risk assessment in the European Union. Trends Biotechnol. 2022 Oct;40(10):1137-1140.
  4. Komala MG, Ong SG, Qadri MU, Elshafie LM, Pollock CA, Saad S. Investigating the Regulatory Process, Safety, Efficacy and Product Transparency for Nutraceuticals in the USA, Europe and Australia. Foods. 2023 Jan 16;12(2):427.
  5. https://www.efsa.europa.eu/en/topics/topic/botanicals
  6. 41. https://ec.europa.eu/commission/presscorner/detail/en/MEMO_11_868

Reviewer 2 Report

Comments and Suggestions for Authors

The paper is well written and informative. It could benefit from additional sub-headings and transitional sentences to help guide the reader through the narrative. The current form is heavy on reciting regulatory text and could benefit from more analysis/synthesis, including comparison with the previous regulatory regime. But in total, this is an informative article and worthwhile contribution to the scholarship.

Author Response

AUTHOR'S REPLY TO THE REVIEW REPORT (REVIEWER 2)

TEXT OF THE ARTICLE

We have added additional subheadings and transitional sentences to guide the reader throughout the narrative.

ANALYSIS/SYNTHESIS

We have carried out further analysis/synthesis on the text.